# Effects of Supplementing Tributyrin on Meat Quality Characteristics of Foreshank Muscle of Weaned Small-Tailed Han Sheep Lambs

**DOI:** 10.3390/ani14081235

**Published:** 2024-04-19

**Authors:** Xue-Er Wang, Zhi-Wei Li, Li-Lin Liu, Qing-Chang Ren

**Affiliations:** 1College of Animal Science and Technology, Tarim University, Alae 843300, China; wangxer_x@163.com; 2College of Animal Science, Anhui Science and Technology University, Chuzhou 233100, China; lizw5440@163.com; 3Anhui Province Key Laboratory of Animal Nutritional Regulation and Health, Anhui Science and Technology University, Chuzhou 233100, China

**Keywords:** tributyrin, amino acid, fatty acid, foreshank muscle, weaned lamb

## Abstract

**Simple Summary:**

Tributyrin has been used as a feed additive to modify the gastrointestinal microflora structure of ruminants, whereas its potential impact on meat quality characteristics has not been well characterized. To fully investigate its potential, an experiment was performed to evaluate the effects of supplementary tributyrin on the meat quality characteristics of foreshank muscle of weaned lambs. When tributyrin was added into the diets of weaned lambs, we found that the meat quality characteristics of foreshank muscle, such as its pH value, redness, water-holding capacity, and intermuscular fat accumulation, as well as its contents of both amino acids and fatty acids, were significantly influenced by supplementing tributyrin. The current research related to the effects of supplementing tributyrin on meat quality characteristics of foreshank muscle of weaned lambs could help with the intensification of nutritional strategies for tributyrin-based livestock systems, improving the meat quality.

**Abstract:**

This experiment aimed to evaluate the effects of supplementing tributyrin (TB) on the meat quality characteristics of foreshank muscle of weaned lambs. A total of 30 healthy weaned Small-Tailed Han female lambs with body weights ranging from 23.4 to 31.6 kg were selected and randomly divided into five groups, and each group consisted of 6 lambs. The control group was fed a basic total mixed ration, while other groups were fed the same ration supplemented with 0.5, 1.0, 2.0, and 4.0 g/kg TB, respectively. The experiment lasted 75 d, including 15 d of adaptation. Foreshank muscle obtained at the same position from each lamb was used for chemical analysis and sensory evaluation. The results showed that supplementing TB increased the muscle contents of ether extract (*p* = 0.029), calcium (*p* = 0.030), phosphorus (*p* = 0.007), and intermuscular fat length (*p* = 0.022). Besides, TB increased the muscle pH (*p* = 0.001) and redness (*p* < 0.001) but reduced the lightness (*p* < 0.001), drip loss (*p* = 0.029), cooking loss (*p* < 0.001), shear force (*p* = 0.001), hardness (*p* < 0.001), cohesiveness (*p* < 0.001), springiness (*p* < 0.001), gumminess (*p* < 0.001), and chewiness (*p* < 0.001). In addition, TB increased the muscle content of inosine-5′-phosphate (*p* = 0.004). Most importantly, TB increased the muscle contents of essential amino acids (*p* < 0.001). Furthermore, TB increased the saturated fatty acids level in the muscle (*p* < 0.001) while decreasing the unsaturated fatty acids content (*p* < 0.001). In conclusion, supplementing TB could influence the meat quality of foreshank muscle of weaned lambs by modifying the amino acid and fatty acid levels.

## 1. Introduction

Small ruminants such as sheep and goats, particularly native breeds, play a significant role in the livelihoods of a considerable part of the human population in the tropics, from a socio-economic perspective [1,2,3]. For example, the Small-Tailed Han sheep is one of the best local breeds recorded in the List of Livestock and Poultry Breed Resources in China, and is famous for its high fecundity, often having 2 L per year or 3 L every 2 years [4]. In addition, the Small-Tailed Han sheep are also renowned for their strong adaptability, stress resistance, and tolerance of low-quality feed [5]. However, this sheep has obvious deficiencies in productive performance and meat quality compared with foreign sheep breeds. Thus, combined trials with emphasis on administration and genetic progress to improve this breed’s meat quality are of decisive significance. However, with marked differences compared to monogastric animals, the fatty acids (**FAs**) composition of ruminant meat is influenced by many factors. The unsaturated fatty acids, particularly polyunsaturated fatty acids, in ruminant meat are significantly influenced not only by diet, but also by the microbial biohydrogenation of dietary lipid in the rumen [6]. The rumen microbes can hydrogenate a large proportion of unsaturated fatty acids so that the FAs in ruminant meat are less unsaturated and more saturated than those in pigs [7]. But the biohydrogenation is incomplete, resulting in some unsaturated fatty acids escaping from the rumen, and following absorption in the small intestine, these are used as substrates for tissue lipogenesis. Thus, the development of the small intestine may also be one of the important factors that affects the nature and the amount of FAs stored in muscle by altering absorption [8]. In these regards, it is possible to modify the FAs composition and provide high-quality mutton for consumers by manipulating dietary nutrition, the biohydrogenation of rumen microbiomes, and intestinal absorption in sheep.

Tributyrin (**TB**) as a green feed additive has recently attracted much attention due to its favorable roles in the production of ruminant animals. For pre-weaned dairy calves, supplementing TB in milk replacer could effectively promote intestinal development by stimulating the colonization of volatile fatty acids-producing bacteria such as *Ruminococcaceae*, *Lachnospiraceae*, *Prevotella*, and *Rikenellaceae* [9]. For weaned Small-Tailed Han sheep, supplementing TB was observed to enhance the relative abundances of rumen bacteria at both the phylum and genus levels [10]. Besides, our previous experiments demonstrated that TB supplementation could significantly increase the yield of microbial crude protein and promote volatile fatty acids concentration in both rumen and fermentation fluids [11,12,13,14]. It is well known that the rumen microbial composition plays a vital role in both the growth and production of ruminant animals through its association with rumen bacteria; thus, we hypothesized that adding TB to the diet had a good potential to improve the meat quality characteristics such as amino acid (**AA**) content and FA level in mutton. Here, our work was carried out to evaluate the effects of TB on the meat quality characteristics of foreshank muscle of weaned Small-Tailed Han sheep. The obtained results could not only fill a gap in the international literature related to the effects of TB on meat quality characteristics but also may provide a scientific basis for TB as a feed additive in the diet of small ruminants.

## 2. Materials and Methods

### 2.1. Sampling Statement

The sampling procedures were applied according to Chinese Guideline no. 398 on the Ethical Treatment of Experimental Animals (2006). According to the cutting order, the mutton could be divided into individual parts such as forequarter, rack, loin, rump-bone, leg-chump, hindshank, breast, and lap, neck and foreshank. Each piece of muscle was strictly trimmed according to the dressing instructions before sampling.

### 2.2. Lambs and the Experimental Design

In order to facilitate management and to eliminate the fixed effect of gender, female Small-Tailed Han sheep were selected for inclusion in the study. A total of 30 healthy 3-month-old weaned female lambs with body weights of 27.5 ± 4.1 kg (mean ± SD) were randomly assigned to 5 groups of 6 lambs each, and each lamb was, respectively, kept in an individual metabolism cage (2.25 m^2^ per cage). The study consisted of 15 d of adaptation to the diet, followed by another 60 d feeding period. During the study, lambs had ad libitum access to water as well as the diet. Each day, the total mixed ration (Table 1) was, respectively, fed at 07:00 and 19:00. Meanwhile, each group of lambs received TB (Perstorp, Shanghai city, China) at levels of 0, 0.5, 1.0, 2.0, and 4.0 g/kg of dry matter (**DM**), and the selected dosage of TB was administered according to the method described by Ren et al. [12], in which TB with different dosages from 0 to 8 g/kg of DM was added into the diet of adult Small-Tailed Han ewes.

### 2.3. Samples Collection

At end of the experiment, all lambs were sacrificed 3 h after the morning feeding according to local slaughtering culture as well as animal welfare standards, and the foreshank muscle was excised from the both forelegs of each lamb according to NY/T1564-2007 [15] (National food safety standard-Cutting technical specification of mutton). In the current experiment, the foreshank muscle was defined as the distal parts derived from the ulna, radius, carpal, and humerus bones and the related muscle, which lies between the elbow and wrist joints. Four small muscles of about 10 g each were, respectively, stored at −20 °C in vacuum packages for intramuscular fat extraction, AAs hydrolysis, nucleotides content determination, and nutrients analysis. The fresh muscle was bloomed in the air for 20 min at 4 °C to determine the meat color, while the shear force, water-holding capacity, and meat texture were measured using muscle samples collected from foreshank after 48 h post-mortem at 4 °C.

### 2.4. Chemical Analysis

The contents of DM, crude protein, ether extract, ash, calcium, and phosphorus of the foreshank muscle and feed were determined according to AOAC methods [16]. According to NRC [17], the metabolizable energy of the feed was estimated. For the determination of both the length and width of intramuscular fat, the muscle sample was sliced into slices of 0.5 mm thickness and sealed with a coverslip (18.0 mm × 18.0 mm) in the center of the slide (25.4 mm × 76.2 mm). A Type C 1 objective lens measuring 1/100 micrometer (Beijing Jiangfengjia Strength Supplier, Beijing, China) was used to correct the Znx-5 Eyepiece micrometer (Dongguan Zhunna Optoelectronic Technology Co., Ltd., Dongguan city, China) at 10 × 10-fold. Thus, the length and width of intramuscular fat in the muscle were determined using a Motic BA210 bio-optical microscope (Macaudie industrial group Co., Ltd., Fujian, Xiamen, China) under the same fold with the Znx-5 Eyepiece micrometer.

### 2.5. Determinations of Muscle pH, Color, Water-Holding Capacity, and Texture

The pH value of the muscle was measured after 15 min postmortem with a portable instrument (HI8424, Beijing Hanna Instruments Science & Technology Co., Ltd., Beijing, China). Muscle color was measured using a colorimeter (Spectrophotometer, CR-10, Minolta, Tokyo, Japan) perpendicular to the muscle surface and the outcome was expressed as L* (lightness), a* (redness), and b* (yellowness). According to our previous methods [18], the water-holding capacity of the foreshank muscle was estimated based on the drip loss and cooking loss, while the texture evaluation (e.g., hardness, cohesiveness, springiness, gumminess, and chewiness) was determined and repeated in triplicates for the samples from each side of the foreshank muscle using a texture analyzer (Model A-XT2, Stable Micro Systems, Surrey, UK). The meat samples of the foreshank were cooked until the internal temperature reached 70 °C, then the cooked samples were cooled to room temperature for shear force determination using a digital muscle tenderness meter (C-LM3B, Northeast Agricultural University, Haerbin, China).

### 2.6. Analysis of AAs Content

The AAs content of the muscle was analyzed according to the methods described by GB 5009.124-2016 [19] (National food safety standard—Determination of amino acid in foods). Briefly, 1.0 g sample of the foreshank muscle was first ground using a mortar, then approximately 0.5 g of the grinded muscle was weighted and hydrolyzed in a vacuum environment with 15 mL 6.0 mol/L hydrochloric acid solution (Guaranteed reagent, Sinopharm Chemical Reagent Co., Ltd., Shanghai, China) for 36 h at 110 °C. After cooling, 1.0 mL of the hydrolysate was mixed with 1.0 mL water, then the mixed solution was dried with nitrogen gas at 50 °C. After drying, the residue was dissolved with 2.0 mL sodium citrate buffer purchased from Sigma-Aldrich (St. Louis, MO, USA) at pH 2.2. The dissolved solution was filtered using 0.22 μm organic filter membrane (Jiete Bio-filtration Co., Ltd., Guangzhou city, China), and 1.0 μL of the filtered solution was used to determine the AAs composition each time.

During the determination, the UPLC–Orbitrap-MS system was implemented using the UHPLC system (Vanquish, Thermo Fisher Scientific, Waltham, MA, USA), which was equipped with a Waters BEH C18 column (50 m length, inside diameter 2.1 mm, particle size 1.7 μm), and the column temperature was set at 55 °C, while the flow rate was 500 μL per min. The mobile phase system consisted of 1.0 mL/L formic acid (purity 98%, Sigma-Aldrich) in water comprised the A phase, whereas the B phase was prepared using 1.0 mL/L formic acid in acetonitrile (purity 99.8%, Sigma-Aldrich). The liquid phase gradient was set as follows: from 0 to 5.5 min, A:B was set at 95:5 (*v*/*v*); 5.5–7.5 min, the ratio of A to B was changed to 75:25 (*v*/*v*); 7.5–9.5 min, the ratio of A to B was decreased to 40:60 (*v*/*v*); 9.5–13.0 min, A to B backed to the original ratio of 95:5 (*v*/*v*). In our analysis, the Q-Exactive Orbitrap mass spectrometer (UHLC-Q-Exactive MS), equipped with a heated ESI source (Thermo Fisher Scientific, Waltham, MA, USA) was used to record the HRMS data. The ESI conditions were set as follows: the temperatures of both capillary and aux gas were separately set at 300 °C and 350 °C; while the gas pressures of sheath, aux and sweep were 40, 10, and 0 arb; the spray voltage was 3000 v. Lastly, data were collected to analyze the effects of TB on AAs content.

### 2.7. Analysis of FAs Composition

In the present experiment, muscle saturated fatty acids, including C4:0, C10:0, C12:0, C13:0, C14:0, C15:0, C16:0, C17:0, C18:0, C20:0, C21:0, C22:0, and C23:0; monounsaturated fatty acids, including C14:1, C15:1, C16:1, C17:1, C20:1, C18:1n9t, C18:1n9c, and C24:1; polyunsaturated fatty acids, including C18:2n6t, C18:2n6c, C18:3n6, C20:2, C18:3n3, C18:3n6, and C22:6n3 were analyzed according to the methods described by GB 5009.168-2016 [20] (National food safety standard—Determination of fatty acids in foods). Based on the FA composition, atherogenicity index = (12:0 + 4 × 14:0 + 16:0)/(monounsaturated fatty acid + polyunsaturated fatty acids), and thrombogenicity index = (12:0 + 16:0 + 18:0)/[(0.5 × monounsaturated fatty acid) + (0.5 × n-6 polyunsaturated fatty acids) + (3 × n-3 polyunsaturated fatty acids) + (n-3 polyunsaturated fatty acids/n-6 polyunsaturated fatty acids)] were calculated according to the method described by Ulbricht and Southgate [21], in which C12:0, C14:0, C16:0, and C18:0 were lauric acid, myristic acid, palmitic acid and stearic acid, respectively. Briefly, the lipids of the foreshank muscle were extracted with chloroform–methanol mixed solution (2:1, *v*/*v*; Thermo Scientific). Before extraction, about 1.0 g foreshank muscle was mixed with 5 mL liquid nitrogen (Anhui Nanhuan, Bengbu city, China) to grind for 10 min, then 20 mg of the ground foreshank muscle was weighted into a 15 mL glass tube and mixed with 2 mL the chloroform–methanol solution for 30 min. The supernatant was methyl esterified in water for 2 h at 80 °C with 5 mL 1.0 mL/L solution of methanol-2% sulfuric acid (*v*/*v*; Merck, Darmstadt, Germany). Lastly, the mixture was extracted with 2 mL of n-hexane purchased from Merck and washed with 5 mL of pure water. The supernatant was collected and stored at 4 °C to analyze the composition of FAs in the muscle.

In the determination, Agilent HP6890 (Agilent Technologies, Santa Clara, CA, USA) equipped with a RT-2560 quartz capillary column (100 m × 250 μm × 0.20 μm) was used to determine the FAs content in 1.0 μL of the supernatant each time. The chromatographic model was set as follows: 0–13.0 min, the initial oven temperature was set at 100 °C; 13.0–21.0 min, the temperature was set up to 180 °C and was maintained for 6 min; then the temperature increased to 230 °C within 20 min. Lastly, the temperature was increased to and maintained at 280 °C. Nitrogen was used as the carrier gas and the split ratio was set as 100:1. Fatty acid peaks were determined using mixed standard retention times, and individual FA was quantified according to its peak area. In order to ensure the units of the determined FA could be compared with human nutritional guidelines, the present levels of FAs in foreshank muscle were expressed as mg/100 g foreshank muscle.

### 2.8. Determination of Nucleotides Content

Inosine-5′-phosphate and guanosine-5′-monophosphate contents of the foreshank muscle were analyzed according to the methods described by GB 5413.40-2016 [22] (National food safety standard—Determination of nucleotide in foods and milk products for infants and young children). About 5 g foreshank muscle was sliced into samples of 1 mm thickness, freeze-dried in a vacuum at −80 °C for 24 h, then ground into powder. Then, 1.0 g of the freeze-dried powder was put into a 50 mL centrifuge tube and 10 mL and 5.0 mL/L trichloroacetic acid were added (Sigma-Aldrich, USA). After 5 min vortex mixing, the mixture was centrifuged with 4000× *g* for 15 min, then the supernatant was filtered with a 0.22 μm filter membrane, and the filtered solution was stored at 4 °C until analysis. Meanwhile, inosine (97%) and guanylate (98%) standard samples purchased from Nanjing Beiyu Biotechnology Co., Ltd. (Nanjing, Jiangsu province, China) were, respectively, weighted as 1.0 ± 0.01 mg, and were dissolved in 1 mL of water to prepare 1 mg/mL inosine and guanylate solution. The solution was diluted to 1.0, 5.0, 25.0, and 50.0 μg/mL inosine and guanylate standard solutions using deionized water.

GC–MS analysis was carried out by Agilent 7890B/7000C (Agilent Technologies, Santa Clara, CA, USA) with a shim-pack C18 column (2.1 mm × 100 mm, 1.8 μm particle size), and the column temperature was set at 40 °C. Mobile phase A was pure acetonitrile, while mobile phase B was prepared with 0.25 mol/L ammonium acetate solution (consisting of 1.0 mL/L formic acid). From 0 to 1.0 min, the ratio of A:B was 10:90; at 1.1–4.0 min, the ratio was increased to 100:0 and maintained for 2 min; from 6.1 to 8.0 min, the ratio decreased to 10:90 again and was also maintained for 2 min. The flow rate was 0.25 mL per min and 1.0 μL of the stored solution was injected each time.

### 2.9. Calculation and Statistical Analysis

In the present study, SAS 9.4 (Statistical Analysis for Windows, SAS Institute Inc., Cary, NC, USA) with a PROC MIXED model was used to analyze the experimental data. The effects of supplementing TB were evaluated using Contrast (0 vs. TB), Linear, and Quadratic effects. Meanwhile, the significance level of the comparison among treatments mean was determined using Duncan’s multiple range test. In our experiment, the difference among the treatment means was considered significant if the *p* < 0.05. The PROC MIXED model was used as follows:*Y_ij_* = *μ* + *L_i_* + *T_j_* + *ε_ij_*
where *Y_ij_* is the dependent variable, *μ* is the overall mean, *L_i_* is the random effects of lambs (*i* = 6), *T_j_* is the fixed effects of supplementing TB (*j* = 0, 0.5, 1.0, 2.0 and 4.0 g/kg), and *ε_ij_* is the error term.

## 3. Results

### 3.1. Effects of TB on Muscle Chemicals, pH, Color, Water-Holding Capacity, and Texture

As shown in Table 2, dietary supplementation with TB increased the contents of ether extract (*p* = 0.029), calcium (*p* = 0.030), phosphorus (*p* = 0.007), and intermuscular fat length (*p* = 0.022) in foreshank muscle. As shown in Table 3, TB increased the muscle pH value (*p* = 0.001) and redness (*p* < 0.001), but TB decreased the muscle lightness (*p* < 0.001), drip loss (*p* = 0.029), cooking loss (*p* < 0.001), as well as the shear force (*p* = 0.001). As shown in Table 4, the muscle hardness (*p* < 0.001), cohesiveness (*p* < 0.001), springiness (*p* < 0.001), gumminess (*p* < 0.001), and chewiness (*p* < 0.001) were decreased by supplementing TB.

### 3.2. Effects of TB on AAs Content of Foreshank Muscle

As shown in Figure 1, the foreshank muscle of lambs fed TB had higher contents of essential amino acids (*p* < 0.001), including valine (*p* < 0.001), methionine (*p* < 0.001), isoleucine (*p* < 0.001), leucine (*p* < 0.001), phenylalanine (*p* = 0.004), and lysine (*p* = 0.014). As shown in Figure 2, TB increased the muscle contents of non-essential amino acids (*p* < 0.001), including serine (*p* = 0.001), proline (*p* < 0.001), glutamic acid (*p* = 0.002), histidine (*p* < 0.001), arginine (*p* < 0.001), and aspartic acid (*p* < 0.001). Therefore, TB increased the muscle contents of ΣAAs (*p* < 0.001), branched-chain AAs (*p* < 0.001), umami (*p* < 0.001), and sweet AAs (*p* < 0.001). As shown in Appendix A, the muscle of lambs fed TB had both higher ratios of essential amino acids to ΣAAs (*p* = 0.003) and non-essential amino acids (*p* = 0.003), but lower ratio of SAA to ΣAAs (*p* < 0.001) compared to the control group. 

### 3.3. Effects of TB on FA Content of Foreshank Muscle

As shown in Figure 3, the foreshank muscle of lambs fed TB had higher content of saturated fatty acids (*p* < 0.001), including C10:0 (*p* = 0.036), C12:0 (*p* = 0.044), C13:0 (*p* < 0.001), C14:0 (*p* < 0.001), C15:0 (*p* < 0.001), C16:0 (*p* = 0.001), C17:0 (*p* = 0.001), and C18:0 (*p* = 0.022). As shown in Figure 4, the muscle of lambs fed TB had lower levels of unsaturated fatty acids (*p* < 0.001) such as C15:1 (*p* < 0.001), C16:1 (*p* < 0.001), C17:1 (*p* < 0.001), C20:1n9 (*p* < 0.001), C18:1n9t (*p* < 0.001), C18:1n9c (*p* < 0.001), C18:2n6t (*p* < 0.001), C18:2n6c (*p* = 0.008), C18:3n3 (*p* = 0.002), C20:3n6 (*p* < 0.001), C20:5n3 (*p* = 0.025), and C22:6n3 (*p* = 0.001). Meanwhile, the muscle of lambs fed TB had lower levels of both monounsaturated fatty acids (*p* < 0.001) and polyunsaturated fatty acids (*p* < 0.001). With the increase in TB, the ratios of both monounsaturated fatty acids (*p* < 0.001) and polyunsaturated fatty acids (*p* < 0.001) to saturated fatty acids linearly decreased. Compared to the control, the muscle of lambs fed TB had lower levels of n3 (*p* < 0.001) and n6 polyunsaturated fatty acids (*p* < 0.001). As shown in Appendix A, the atherogenicity index (*p* < 0.001) and thrombogenicity index (*p* < 0.001) linearly increased with increasing TB.

### 3.4. Effects of TB on Nucleotides Content

As shown in Figure 5, the foreshank muscle of lambs fed TB at a dose of 4.0 g/kg DM had higher inosine-5′-phosphate content (*p* = 0.004) but lower guanosine-5′-monophosphate content (*p* = 0.003) compared with lambs in the control group.

## 4. Discussion

Limited research determining effects of supplementing TB on the meat quality characteristics of weaned lambs is currently available, and this will help with the intensification of nutritional strategies of TB-based livestock systems in improving meat quality. The present study showed that supplementing TB affected the meat quality characteristics such as fat accumulation, muscle pH, color, water-holding capacity and texture as well as both AA and FA levels of foreshank muscle in weaned Small-Tailed Han sheep.

### 4.1. Effects of TB on Intramuscular Fat Accumulation in Foreshank Muscle

In our experiment, supplementing TB could significantly increase the content of ether extract and intramuscular fat length in foreshank muscle, and this may be associated with the stimulating effects of TB on regulating factors related to lipid transcription, absorption, and metabolism. Our unpublished data, together with the experimental results of Xiong et al. [23], demonstrated that dietary addition of TB could increase the serum levels of high-density lipoprotein cholesterol, which is well known due to its function in transferring peripheral blood cholesterol to the liver. In the liver, cholesterol can be used to synthesize bile acids, which can, in turn, be secreted into the small intestine to stimulate lipid absorption [24]. Besides, Liu et al. [9] and Li et al. [10] reported that supplementing TB could successfully enhance small intestine developments of both dairy cows and weaned lambs, which would also be beneficial for lipid absorption. Furthermore, a recent study by Gu et al. [25] pointed out that TB supplementation at dosage of 1.0 g/kg in basal diet not only increased the serum concentration of high-density lipoprotein cholesterol but also upregulated the lipid metabolism-related gene expression levels of *peroxisome proliferators-activated receptor*, *proliferators-activated receptor*, *fatty acid synthase*, *lipoprotein lipase*, and *adipose triglyceride lipase* in the liver of broilers, which resulted in an increase in fat deposition by promoting fat synthesis. Intramuscular fat accumulation is a very complex process; thus, more research is required to determine if a similar effect of TB on upregulating gene expression related to lipid metabolism observed in broilers can be also achieved in weaned lambs, since our experiment showed a positive effect of TB on fat accumulation in foreshank muscle.

### 4.2. Effects of TB on Nutritional Composition, pH, Color, Water-Holding Capacity, and Texture in Foreshank Muscle

In the current study, TB could increase the contents of calcium and phosphorus in foreshank muscle, and this may be associated with the TB improving the development of small intestine, which is the major absorption site of both calcium and phosphorus [26]. Our previous study demonstrated that TB could significantly decrease daily excretions of calcium and phosphorus in the feces and urine of Small-Tailed Han ewes [12]. Recently, Li et al. reported that supplementary TB could effectively improve the development of the small intestine of weaned lambs [10], which could contribute to absorptions of both calcium and phosphorus, thus resulting in higher contents of calcium and phosphorus in foreshank muscle.

In this experiment, the foreshank muscle of lambs fed diets with supplementation of TB had higher pH values, and this may be due to the inhibiting effect of TB on lactate dehydrogenase activity. Our unpublished data showed that lambs fed TB had lower lactate dehydrogenase activity, which was beneficial for reducing the decline in pH in foreshank muscle. Meat quality characteristics, including muscle color, water-holding capacity, and shear force, are influenced by many factors, such as postmortem aging and the antioxidative stability of muscle [27]. In the present study, TB was observed to markedly elevate foreshank muscle redness of lambs, and this may be associated with the antioxidative function of TB in meat. At present, little is known about the benefit of TB on the antioxidative function in sheep, but some information about the effect of TB can be extrapolated from studies in simple-stomached species. Accordingly, Wang et al. [28] reported that dietary supplementation with 1.0 g/kg TB significantly enhanced the total antioxidant capacity of the ovaries in broilers. Besides, Hu et al. [29] found that supplementing TB at a dosage of 2.0 g/kg in feed could improve the red blood cell count of broilers at 21 d. However, possible species differences have to be taken into account.

The present study showed that TB could also improve the eating quality of foreshank muscle in lambs, such as the hardness, cohesiveness, springiness, gumminess, and chewiness, and this may be due to the enhancing effects of TB on water-holding capacity in foreshank muscle. A study by Yu et al. [30] demonstrated that meat texture characteristics such as hardness, cohesiveness, springiness, gumminess and chewiness in Tan sheep were negatively correlated with the water-holding capacity. Based on the present results, supplementing TB as an effective feed additive had good potential to improve the eating quality of foreshank muscle.

### 4.3. Effects of TB on AAs and Nucleotides Content in Foreshank Muscle

In the present experiment, supplementing TB could increase the contents of essential amino acids in foreshank muscle, and this may be associated with positive effects of TB on microbial crude protein synthesis and small intestine development. Ruminants exhibit marked differences to monogastric animals such as pigs. Ruminants’ protein sources consist of three parts: a small amount of endogenous protein derives from cast-off cells, an important part of the protein required by ruminants derives from dietary crude protein, while most of the rest is contributed by microbial crude protein synthesized in the rumen. When dietary crude protein is taken into the rumen, ruminants consume part of the crude protein to obtain AAs for microbial crude protein synthesis. The synthesized microbial crude protein, together with the dietary crude protein that has escaped from the rumen degradation, and the endogenous protein pass through the small intestine to meet the requirements of ruminants by contributing AAs [31]. Among the AAs required by ruminants, microbial crude protein derived from the rumen is the most notable AAs source because it can provide approximately 810 g/kg of the qualitative AA requirements of growing lambs [32]. Furthermore, the microbial crude protein synthesized in the rumen is an excellent protein source because it has a relatively good AA balance and digestibility compared with dietary crude protein [33]. Sok et al. [34] reported that there were at least 18 types of AAs in the rumen bacteria and protozoa, including the 7 essential amino acids detected in the current experiment. Our previous study found that TB could effectively increase the ruminal yield of microbial crude protein in Small-Tailed Han sheep [11]. Thus, the elevated microbial crude protein reaching the small intestine represents not only the greater contribution of protein for sheep but also the increasing essential amino acids content in foreshank muscle. In addition, TB has been demonstrated to stimulate the small intestinal development of weaned lambs [10], which is also beneficial for AAs absorption. According to FAO [35], for healthy food, the ratios of essential amino acids to both total AA and non-essential amino acids at least are 0.40 and 0.60, respectively. In the present study, the average ratio of essential amino acid to total AA was approximately 0.42 while the average ratio of essential amino acid to non-essential amino acid was approximately 0.74 in TB treatments, and this indicated that dietary supplementations with TB could promote the healthy status of the foreshank muscle by improving the AAs composition, particularly the essential amino acids content.

A previous study showed that inosine-5′-phosphate is responsible for the umami taste of meat, and the more inosine-5′-phosphate the meats of animals contain, the better they taste [36]. But the inosine-5′-phosphate content in the meat varies with individual samples, and is affected by many factors, such as genes, the animal breed, age, sex, feed, tissue position, cooking conditions, and so on [37]. The present study shows that TB as a feed additive could also affect the accumulations of inosine-5′-phosphate and guanosine-5′-monophosphate in the foreshank muscle of lambs, and this may be associated with the antioxidative effects of TB. With bio-reactions of metabolic enzymes in muscle, adenosine triphosphate can be degraded to adenosine diphosphate after slaughter, following generations of adenosine monophosphate and inosine monophosphate [36]. The generated inosine monophosphate can be further catalyzed into xanthosine monophosphate by inosine monophosphate dehydrogenase, followed by production of guanosine-5′-monophosphate [38]. Van der Knaap and Verrijzer [39] reported that the inosine monophosphate dehydrogenase can accumulate in response to oxidative or replicative stress. As described in previous work, TB has been shown to effectively enhance the total antioxidant capacity by improving the activity of antioxidative enzymes [28]. Thus, dietary supplementation of TB may reduce the accumulation of the inosine monophosphate dehydrogenase, which could result in more contents of inosine monophosphate escaping through hydrolysis of the dehydrogenase and lower contents of guanosine-5′-monophosphate being generated in the muscle. The present study indicates that supplementing TB could improve the umami taste of mutton by increasing the accumulation of inosine-5′-phosphate in foreshank muscle.

### 4.4. Effects of TB on FAs Level in Foreshank Muscle

In recent years, more and more studies have showed that the rumen microbiome is an important factor influencing FA composition, particularly the relative proportions of saturated fatty acid, monounsaturated fatty acids, and polyunsaturated fatty acids in meat derived from ruminants. Xiong et al. [40] found that the content of n3-polyunsaturated fatty acids in the foreshank muscle of Hu lambs was positively correlated with the ruminal *Christensenellaceae_R-7_group*. Besides, Zhang et al. [41] used a metabolomics approach to reveal that the muscle content of linoleic acid in *longissimus lumborum* of Black Tibetan sheep was positively correlated with the abundances of *Quinella*, *Ruminococcus 2* and *coprostanoligenes* (*Eubacterium*) in the rumen. In the present study, dietary supplementation with TB could change the composition of FAs in the foreshank muscle of weaned Small-Tailed Han lambs by increasing the content of saturated fatty acids in the muscle while decreasing the amount of unsaturated fatty acids, including monounsaturated fatty acids and polyunsaturated fatty acids, and this may be due to the stimulating effects of TB on rumen microbes responsible for biohydrogenation. Potu et al. [42] reported that fibrolytic bacteria such as *Fibrobacter*, *Ruminococcus,* and *Butyrivibrio* are important in the biohydrogenation process of dietary unsaturated fatty acids. For example, *Fibrobacter* could convert C15:1 to C15:0 [43], while *Butyrivibrio* can bio-hydrogenate linoleic acid (C18:2n6c) to form C18:0 [44]. Also, Boeckaert et al. [45] reported that *Butyrivibrio* is the principal rumen bacteria involved in biohydrogenation of C18:1 FA. In addition, Jeyanathan et al. [46] demonstrated that *Butyrivibrio* could also bio-hydrogenate 22:6n-3 into unsaturated 22 carbon FAs between 22:6n-3 and 22:0. Our previous experiment demonstrated that dietary supplementation with TB could stimulate the relative abundance of rumen bacteria such as *Butyrivibrio*, *Streptococcus*, and *Fibrobacter* [10], and this would promote the microbial biohydrogenation of unsaturated fatty acids and the formation of saturated fatty acids in the rumen. Although some of the synthesized saturated fatty acids are commonly considered to be harmful to human health, there are benefits of the saturated fatty acids such as C18:0, C16:0 and C14:0, which can provide more energy value, higher resistance to oxidation, and a greater octane number for better combustion efficiency [47].

Eicosapentaenoic acid (**EPA**) has high health amelioration potentials, hence it is of great interest to increase the EPA content in meat. In the current experiment, TB could decrease the content of EPA in foreshank muscle, and this may be associated with the stimulating effects of TB on the rumen *Clostridium* [10]. Sakurama et al. [48] reported that *Clostridium* could effectively convert both arachidonic acid (*cis*-5, *cis*-8, *cis*-11, *cis*-14-eicosatetraenoic acid) and EPA (*cis*-5, *cis*-8, *cis*-11, *cis*-14, *cis*-17-EPA) into *cis*-5, *cis*-8, *trans*-13-eicosatrienoic acid and *cis*-5, *cis*-8, *trans*-13, *cis*-17-eicosatetraenoic acid. Thus, TB could decrease the level of EPA in foreshank muscle of lambs, which was not beneficial for promoting the meat quality. The atherogenicity index and thrombogenicity index are related to the profile of FAs with the risk of cardiovascular disease, which could be decreased by the content of unsaturated fatty acids, particularly polyunsaturated fatty acids [21]. Since TB could effectively increase the level of saturated fatty acids, there were increases in the atherogenicity index and thrombogenicity index in TB treatments.

## 5. Conclusions

The present experiment shows that supplementing TB could promote pH value, redness, and water-holding capacity, as well as intermuscular fat accumulation in the foreshank muscle of weaned lambs. Besides, TB increased the content of inosine-5′-phosphate in the muscle. However, TB decreased the muscle shear force and texture. Most importantly, TB could increase the muscle content of essential amino acids. Furthermore, TB could modify the muscle FAs composition by increasing the level of saturated FAs while decreasing the level of unsaturated FAs. The above results indicate that supplementing TB could influence the meat quality characteristics of the foreshank muscle of lambs by modifying the muscle contents of both AAs and FAs.

## Figures and Tables

**Figure 1 animals-14-01235-f001:**
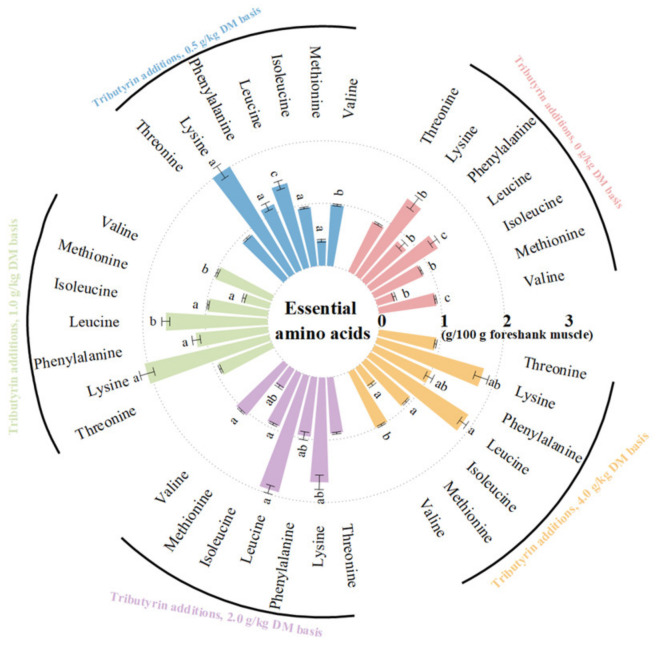
The effects of tributyrin on the contents of essential amino acids in the foreshank muscle of weaned lambs. ^a–c^ Values within an essential amino acid with no common superscripts differ significantly (*p* < 0.05).

**Figure 2 animals-14-01235-f002:**
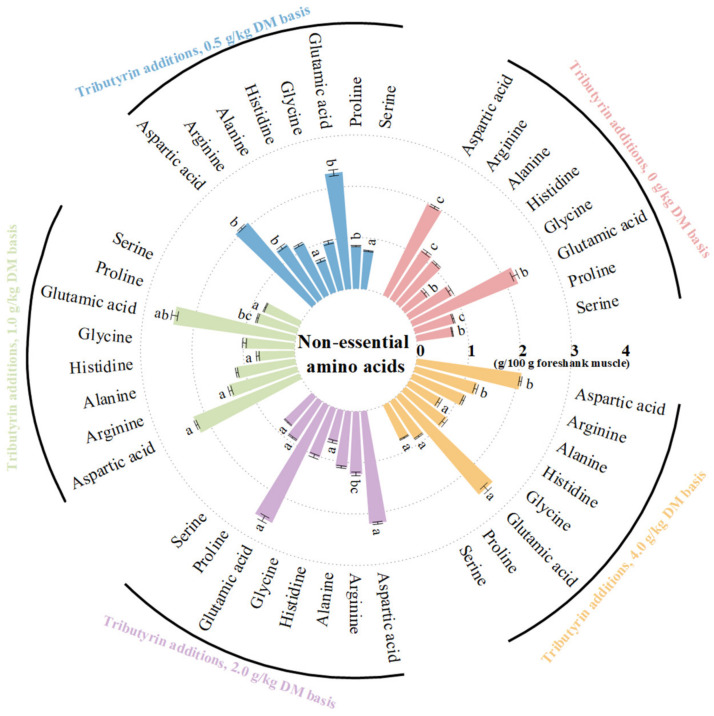
The effects of tributyrin on contents of non-essential amino acids in foreshank muscle of weaned lambs. ^a–c^ Values within a non-essential amino acid with no common superscripts differ significantly (*p* < 0.05).

**Figure 3 animals-14-01235-f003:**
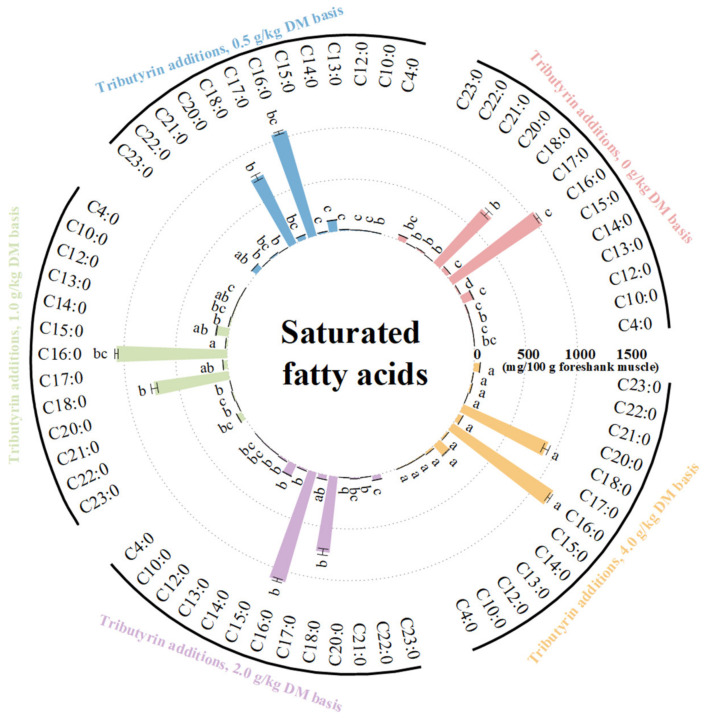
The effects of tributyrin on contents of saturated fatty acids in foreshank muscle of weaned lambs. ^a–d^ Values within a saturated fatty acid with no common superscripts differ significantly (*p* < 0.05).

**Figure 4 animals-14-01235-f004:**
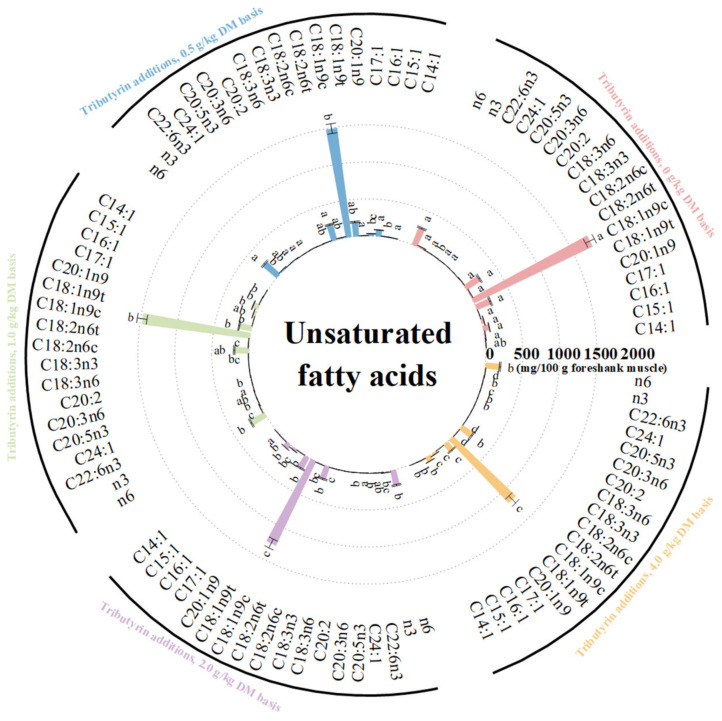
The effects of tributyrin on contents of unsaturated fatty acids in foreshank muscle of weaned lambs. ^a–d^ Values within an unsaturated fatty acid with no common superscripts differ significantly (*p* < 0.05).

**Figure 5 animals-14-01235-f005:**
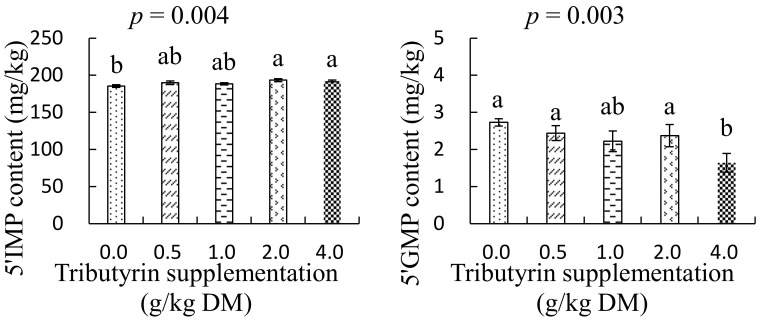
The effects of tributyrin on contents of inosine-5′-phosphate (5′-IMP) and guanosine-5′-monophosphate (5′-GMP) in foreshank muscle of Small-Tailed Han lambs. ^a–b^ Values within diferent supplementations with no common superscripts differ significantly (*p* < 0.05).

**Table 1 animals-14-01235-t001:** Ingredient and nutrient composition of the total mixed ration feed for weaned Small-Tailed Han female lambs (g/kg, as DM basis).

Items	Content
Ingredients	
Maize	250
Soybean meal	110
Ensiled total corn stover	350
Peanut straw	200
Garlic by-products	50.0
Premix ^1^	40.0
Nutrients	
Metabolizable energy ^2^, MJ/kg	12.3
Crude protein	181
Ether extract	31.0
NDF	373
ADF	242
Non-fiber carbohydrate ^3^	341
Ash	74.0
Ca	7.0
Total P	4.0

DM = dry matter; NDF = neutral detergent fiber; ADF = acid detergent fiber. ^1^ Each kilogram of premix contained 154 kIU of vitamin A, 94.0 kIU of vitamin D_3_, 338 kIU of vitamin E, 120 mg of I, 280 mg of Cu, 2240 mg of Fe, 1740 mg of Mn, 1370 mg of Zn, 60.0 mg of Se, 16.8 mg of Co, 50.0 mg of Lys and 50.0 mg of Met. ^2^ Metabolizable energy was based on calculated values. ^3^ Non-fiber carbohydrate (g/kg, DM basis) = 1000 − (NDF + crude protein + ether extract + ash).

**Table 2 animals-14-01235-t002:** The effects of tributyrin on nutritional chemicals and intermuscular fat of foreshank muscle in Small-Tailed Han lambs.

Items	Tributyrin Additions, g/kg DM Basis					SEM	*p*-Values ^1^		
	0	0.5	1.0	2.0	4.0		Contrast	Linear	Quadratic
Dry matter (g/100 g)	25.5	26.3	26.9	26.7	25.3	0.96	0.476	0.980	0.961
Protein (g/100 g)	18.8	19.5	19.7	19.6	18.1	0.99	0.703	0.665	0.845
Ether extract (g/100 g)	5.94 ^b^	6.20 ^ab^	6.67 ^a^	6.53 ^ab^	6.61 ^ab^	0.214	0.029	0.021	0.372
Ash (mg/100 g)	715	625	699	745	676	69.8	0.292	0.510	0.779
Calcium (mg/100 g)	4.98 ^b^	6.61 ^ab^	6.63 ^ab^	8.15 ^a^	6.62 ^ab^	0.786	0.030	0.035	0.254
Phosphorus (mg/100 g)	48.1 ^b^	59.0 ^b^	78.1 ^a^	62.5 ^b^	58.5 ^b^	4.98	0.007	0.137	0.041
Intermuscular fat length (μm)	37.2 ^b^	42.7 ^ab^	44.4 ^ab^	49.1 ^a^	41.9 ^ab^	2.69	0.022	0.045	0.342
Intermuscular fat width (μm)	12.6 ^b^	15.0 ^ab^	12.4 ^b^	17.8 ^a^	15.1 ^ab^	1.56	0.176	0.131	0.033

DM = dry matter; SEM = standard error of the mean. ^a,b^ Values within a row with no common superscripts differ significantly (*p* < 0.05). ^1^ Linear = linear effect of tributyrin; Quadratic = quadratic effect of tributyrin.

**Table 3 animals-14-01235-t003:** The effects of tributyrin on pH, color, and water-holding capacity and shear force in foreshank muscle of Small-Tailed Han lambs.

Items	Tributyrin Additions, g/kg DM Basis					SEM	*p*-Values ^1^		
	0	0.5	1.0	2.0	4.0		Contrast	Linear	Quadratic
pH^2^	6.60 ^b^	6.84 ^ab^	7.02 ^a^	6.72 ^b^	7.09 ^a^	0.085	0.001	0.002	0.033
L* (lightness)	36.4 ^a^	32.7 ^b^	32.1 ^b^	31.1 ^b^	31.4 ^b^	1.00	<0.001	<0.001	0.532
a* (redness)	14.6 ^c^	18.1 ^b^	20.6 ^a^	17.6 ^b^	16.9 ^bc^	0.84	<0.001	0.130	0.043
b* (yellowness)	3.36	3.43	3.74	2.97	3.09	0.263	0.856	0.235	0.142
Drip loss_24h_ (g/100 g)	7.91 ^a^	4.48 ^b^	6.03 ^ab^	6.47 ^ab^	5.47 ^ab^	0.919	0.029	0.325	0.045
Cooking loss (g/100 g)	31.6 ^a^	28.8 ^ab^	23.1 ^c^	24.5 ^c^	25.9 ^bc^	1.33	<0.001	<0.001	0.119
Shear force (N)	25.5 ^a^	21.1 ^ab^	19.7 ^b^	17.6 ^b^	19.2 ^b^	1.66	0.001	0.004	0.555

DM = dry matter; SEM = standard error of the mean. ^a–c^ Values within a row with no common superscripts differ significantly (*p* < 0.05). ^1^ Linear = linear effect of tributyrin; Quadratic = quadratic effect of tributyrin. ^2^ Muscle pH value was measured at 15 min postmortem.

**Table 4 animals-14-01235-t004:** The effects of tributyrin on foreshank muscle texture of Small-Tailed Han lambs.

Items	Tributyrin Additions, g/kg DM Basis					SEM	*p*-Values ^1^		
	0	0.5	1.0	2.0	4.0		Contrast	Linear	Quadratic
Hardness (g)	228 ^a^	215 ^b^	193 ^c^	183 ^d^	185 ^cd^	3.3	<0.001	<0.001	0.527
Cohesiveness	0.59 ^a^	0.57 ^a^	0.53 ^b^	0.50 ^bc^	0.49 ^c^	0.010	<0.001	<0.001	0.741
Springiness	0.30 ^a^	0.26 ^b^	0.24 ^b^	0.20 ^c^	0.21 ^c^	0.009	<0.001	<0.001	0.164
Gumminess (g)	34.2 ^a^	31.2 ^b^	25.8 ^c^	23.0 ^d^	23.1 ^d^	0.78	<0.001	<0.001	0.495
Chewiness (g)	42.0 ^a^	33.7 ^b^	25.2 ^c^	18.7 ^d^	20.0 ^d^	1.65	<0.001	<0.001	0.762

DM = dry matter; SEM = standard error of the mean. ^a–d^ Values within a row with no common superscripts differ significantly (*p* < 0.05). ^1^ Linear = linear effect of tributyrin; Quadratic = quadratic effect of tributyrin.

## Data Availability

Ante-mortem information about the lambs, such as their growth performance, dry matter intake, dietary nutrients digestibility, and carcass, is available in both Table 2 and Table 3 in our previous report (https://doi.org/10.1016/j.aninu.2023.08.006).

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
