# Peer review of "Effects of Supplementing Tributyrin on Meat Quality Characteristics of Foreshank Muscle of Weaned Small-Tailed Han Sheep Lambs"

_animals, 2024, doi:10.3390/ani14081235_

Round 1
Reviewer 1 Report
Comments and Suggestions for Authors
The article evaluates the effect of Tributyrin supplementation on the quality characteristics of sheep meat. I do not recommend publishing the manuscript as presented as I understand that ante-mortem information (consumption, digestibility, performance and carcass) is necessary. Only the characteristics of the meat offer us partial conclusions about the use of Tributyrin in the animals' diet. I recommend that the manuscript be rewritten and ante-mortem information be added.
Author Response
Dear reviewer 1#,
We highly appreciate your valuable comments on our manuscript. We agree with you that the ante-mortem information such as growth performance, dry matter intake, nutrients digestibility and carcass are necessary. Above necessary data are available in both Table 2 and Table 3 of our previous report (https://doi.org/10.1016/j.aninu.2023.08.006). In this reversion, the information has been added in section of Data Availability Statement. Thanks again!
Kind regards,
Ren, Q. C.

Reviewer 2 Report
Comments and Suggestions for Authors
Effects of Supplementing Tributyrin on Meat Quality Characteristics of Foreshank Muscle of Weaned Small-Tailed Han Sheep lambs.
The manuscript presents original research, which is relevant and of importance to the scientific community. The document is generally well structured and easy to understand. However, I have some observations and recommendations that must be addressed before acceptance.
Line 106-108: Indicate the specific order in which each piece of muscle was used for each type of response variable. It is important
Lines 109-110: Indicate the size of the meat sample for determining color, texture and water holding capacity. Also, indicate the anatomical position (in foreshank muscle) of each piece used in each determination.
Line 125: Why was pH measured at 15 min postmotem and not at 45 min?
Line 130: Describe the variables included in the texture evaluation. Shear force, hardness, cohesiveness, springiness, gumminess, and chewiness?
Line 134: How many replicates per meat sample were made for texture determination?
Section 2.7. It is necessary to indicate that the sums of SFA, MUFA, PUFAs, atherogenicity and thrombogenicity indices were calculated.
Discussion: It would be important to discuss why or how TB causes the observed changes. It only focuses on contrasting the results with previous studies.
Author Response
Dear reviewer 2#,
We highly appreciate your detailed valuable comments on our manuscript. In this corrected manuscript, we carefully attend to and clarify your concerns point by point. And we hope the reviewers will be satisfied with our responses to the comments and the revisions for this paper. Thanks again!
Kind regards,
Ren, Q. C.
The detailed reversion
2.1 Line 106-108: Indicate the specific order in which each piece of muscle was used for each type of response variable. It is important
Au: Accepted. The specific order of cutting mutton has been added in Lines 114-118. Thanks for your valuable suggestion.
2.2 Lines 109-110: Indicate the size of the meat sample for determining color, texture and water holding capacity. Also, indicate the anatomical position (in foreshank muscle) of each piece used in each determination.
Au: Thanks for your suggestions. The sampling size had been indicated in 2.3. Samples collection and 2.9. Calculation and statistical analysis, respectively. Since the number of lambs used in the present study was relatively smaller, all lambs (30) were sacrificed at end. Thus, for each treatment the sampling size of 6 Í2 (left and right) were used to determine color, texture and water holding capacity.
In practice, accurately cutting each piece of foreshank muscle is very difficult, and this neither does not accord with Chinese people's habits of consumption for whole foreshank muscle. But it is important to clarify that the used muscle must be made in the same position from each lamb for chemical analysis and sensory evaluation, which has been emphasized in the abstract.
2.3 Line 125: Why was pH measured at 15 min postmortem and not at 45 min?
Au: Thanks for your suggestions. The current experiment was carried in summer (June to August, called as the festival of sheep), with higher temperature, slaughter experts recommend that the time of acidification should not exceed 20 minutes. Thus the present pH was measured at 15 min postmortem rather than at 45 min. Thanks again.
2.4 Line 130: Describe the variables included in the texture evaluation. Shear force, hardness, cohesiveness, springiness, gumminess, and chewiness?
Au: Accepted. In the present, the texture was evaluated using hardness, cohesiveness, springiness, gumminess and chewiness, which has been described in Line 167. Thanks for your valuable suggestion.
2.5 How many replicates per meat sample were made for texture determination?
Au: In the present study, the measurement was repeated in triplicates for the sample from each side of the foreshank muscle. Thanks for your valuable suggestion.
2.6 Section 2.7. It is necessary to indicate that the sums of SFA, MUFA, PUFAs, atherogenicity and thrombogenicity indices were calculated.
Au: Accepted. The reversion has been added in Lines 205-209 and Lines 211-218, respectively. Thanks for your valuable suggestion.
2.7 Discussion: It would be important to discuss why or how TB causes the observed changes. It only focuses on contrasting the results with previous studies.
Au: Thanks for your suggestions. During the reversion, we did our best to find references related to TB on meat quality characteristics. However, there is little information about how TB causes the observed changes. To fully investigate the potential reasons of the present finding, we continually have been carrying out the relevant research work. Fortunately, the potential nutrient mechanism of TB has been found by our team so far, which displayed its benefits by improving the development of gastrointestinal bacteria rather than via releasing butyric acid. We believe that more effects of TB will be revealed with further research. Thanks again.

Reviewer 3 Report
Comments and Suggestions for Authors
The study aimed to assess effects of tributyrin on meat quality characteristics of lambs, and to clarify the potential for utilization of the substance as a feed additive in diet of lambs.
The authors try to fill a gap in the international literature, but they do not argue correctly for the feasibility of their study. I study to underline the literature gaps better and to try and explain more explicitly where exactly their study can fit.
In the Introduction, please add a new paragraph to described in brief the breed of sheep used in the study. Do you think that the findings can be applicable to other breeds as well? Or are they of local interest only?
At the end of the Introduction, the authors should clarify how they think that their work fits in with other relevant publications and how does it contribute to the international knowledge available.
The authors must make specific improvements in the methodology. For example, please describe how the animals were selected for inclusion into the study. Also, please provide details regarding purveyors of all chemicals and equipment used in the work (there are some omissions in this respect). Also, please justify the use of these controls within the study.
Additional comments about tables and figures.
Some of the tables are very long and are difficult to read. Hence, they can be moved to supplementary material and replaced by corresponding figures and graphs. The figures should be colourised to allow readers to understand the meaning quickly.
In the Discussion, please try to minimise the use of abbreviations, as this is tiresome for readers.
The Conclusions are not fully consistent with the findings, and they should be toned down. Also, do you have a patent for these findings? Do you plan to commercialize the use of tributyrin? The potential clinical advantages and limitations of using it must be added in that section.
Overall: the manuscript needs extensive corrections and re-evaluations followed by a new recommendation.
Author Response
Dear reviewer 3#,
We highly appreciate your detailed valuable comments on our manuscript. In this corrected manuscript, we carefully attend to and clarify the concerns raised by the reviewer. And we hope the reviewers and the editors will be satisfied with our responses to the comments and the revisions for this paper. Thanks again!
Kind regards,
Ren, Q. C.
The detailed reversion
3.1 In the Introduction, please add a new paragraph to describe in brief the breed of sheep used in the study. Do you think that the findings can be applicable to other breeds as well? Or are they of local interest only?
Au: Accepted. The description has been added in Lines 68-74. Thanks for your valuable suggestion. We firmly believe that the benefits of tributyrin can be applicable to other breeds such as Hu sheep as well as dairy calves.
[1] Zhang, X. L., Wang, Z. B., Chen, P. Y., Wang, Y. X., Zhang, Y. L., Wang, F., You, P. H., Fan, Y. X. 2023. Effects of tributyrin supplementation on rumen fermentation indexes and microbial diversity of lambs under negative energy balance in perinatal Hu sheep. https://kns.cnki.net/kcms/detail/32.1148.S.20230406.1740.010.html
[2] Liu, S.; Wu, J.; Wu, Z.; Alugongo, G. M.; Khan, M. Z.; Li, J. H.; Xiao, J. X.; He, Z. Y.; Ma, Y. L.; Lo, S. L.; Cao, Z. J. 2022. Tributyrin administration improves intestinal development and health in pre-weaned dairy calves fed milk replacer. Animal Nutrition 10, 399-411.
3.2 At the end of the Introduction, the authors should clarify how they think that their work fits in with other relevant publications and how does it contribute to the international knowledge available.
Au: Accepted. The importance of our work has been clarified in Lines 99-101 and Lines 105-108. Thanks for your valuable suggestion.
3.3 The authors must make specific improvements in the methodology. For example, please describe how the animals were selected for inclusion into the study. Also, please provide details regarding purveyors of all chemicals and equipment used in the work (there are some omissions in this respect). Also, please justify the use of these controls within the study.
Au: Accepted. The improvements have been listed in Lines 121-122; Lines 180-181; Lines 183-185; Line 188; Line 191; Line 193; Lines 219-221; Lines 224-225; Line 247; Lines 251-252, respectively. Thanks for your valuable suggestions.
3.4 Additional comments about tables and figures.
Some of the tables are very long and are difficult to read. Hence, they can be moved to supplementary material and replaced by corresponding figures and graphs. The figures should be colorized to allow readers to understand the meaning quickly.
Au: Accepted. In order to make the results to be relatively easy understand for readers, we have tried to change the data related to both AAs and FAs into figures despite it was very difficult to do so for many works. Please see the Figure 1 and Figure 2, thanks for your valuable suggestions.
3.5 In the Discussion, please try to minimize the use of abbreviations, as this is tiresome for readers.
Au: Accepted. In this reversion, we have tried to minimize the use of some un-normal abbreviations such as EAA, NEAA, MCP, CP, 5’-IMP, 5’-GMP, ATP, ADP, AMP, IMP, SFA, UFA, MUFA, PUFA, AI and TI. Thanks for your valuable suggestions.
3.6 The Conclusions are not fully consistent with the findings, and they should be toned down. Also, do you have a patent for these findings? Do you plan to commercialize the use of tributyrin? The potential clinical advantages and limitations of using it must be added in that section.
Au: Accepted. The recommend dosage of TB has been deleted in this reversion. Thanks for your valuable suggestions.
To date, the relevant research results are applying for national invention patent in China. However, due to the relatively smaller number of lambs used in the present study, we need to carry out more works to well prove the present findings before the TB are commercially used in practice. At last, we want to share a good news with you that the potential nutrient mechanism of TB has been found by our team, which displayed its benefits by improving the development of gastrointestinal bacteria rather than via releasing butyric acid. Thanks again.

Round 2
Reviewer 3 Report
Comments and Suggestions for Authors
The authors have made substantial changes and have improved the manuscript.
Author Response
Thanks for your encouragement. In this reversion, the Figures 1-4 related to amino acids and fatty acids have been refined with additions of both shoulder letters and error bars, and this will make them to be clear and concise. At last, thanks for your detailed valuable suggestions again on our manuscript.
